# Machine learning for the prediction of urosepsis using electronic health record data

Varuni Sarwal[1], Nadav Rakocz[1], Georgina Dominique[2], Jeffrey N. Chiang[3],
A. Lenore Ackerman [2]*

1 Department of Computer Science, University of California Los Angeles, Los Angeles, California, United States of America, 2 Division of Urogynecology and Reconstructive Pelvic Surgery, Department of Urology, David Geffen School of Medicine at UCLA, Los Angeles, California, United States of America, 3 Department of Computational Medicine, University of California Los Angeles, Los Angeles, California, United States of America

* aackerman@mednet.ucla.edu

## Abstract

Urosepsis, a medical condition resulting from the progression of urinary tract infection (UTI), is a leading cause of death in the US. Urosepsis occurs due to complicated UTI and constitutes ~25% of sepsis cases. Early prediction of urosepsis is critical in providing personalized care, reducing diagnostic uncertainty, lowering mortality rates. While machine learning (ML) techniques have the potential to aid healthcare professionals in identifying risk factors and recommending treatment options, no study has been developed to predict the development of urosepsis in patients with a suspected UTI in outpatient settings. We develop and evaluate ML models in predicting hospital admission and urosepsis diagnosis for patients with an outpatient UTI encounter, leveraging de-identified electronic health records sourced from UCLA. Inclusion criteria included a positive diagnosis of urinary tract infection indicated by ICD-10 code N30/N93.0 and positive bacteria result via urinalysis in an ambulatory setting. W extracted demographic information, urinalysis findings, and antibiotics prescribed for each instance of UTI. Reencounters we defined as encounters within seven days of the initial UTI. Reencounters were considered urosepsis-related if matching positive blood and urine cultures were found with a sepsis ICD-10 code of A41/R78/R65. ML models were trained and evaluated on two tasks: prediction of a reencounter leading to hospitalization, prediction of Urosepsis. Model performances were stratified by ethnicities. Random forest models achieved significant improvement over baseline performance (APR = 0.004), with APR = 0.15 for reencounters and 0.31 for urosepsis prediction. While these APR values reflect the challenge of predicting rare events (0.4% prevalence), they represent meaningful predictive power for clinical risk stratification. We computed Shapley values to interpret model predictions and found patient age, sex, and urinary WBC-count were the top three predictive features. Our study has the potential to assist clinicians in the identification of high-risk patients, informed decisions about antibiotic prescription and improving patient care.

**Data availability statement:** The data that support the findings of this study are publicly available from the University of California, Los Angeles, Health System Discovery Data Repository (https://it.uclahealth.org/about/ohia/products/discovery-data-repository). Unique data utilized in the derivation of the described models can be found at https://doi.org/10.7910/DVN/SUCOFW.

**Funding:** No specific funding supported this work. ALA is supported by the National Institutes of Health (NIDDK K08 DK118176) and the Department of Defense Congressionally Directed Medical Research Programs (W81XWH2110644). The funders had no role in study design, data collection and analysis, decision to publish, or preparation of the manuscript.

**Competing interests:** I have read the journal's policy and the authors of this manuscript have the following competing interests: ALA receives grant funding from Medtronic, Inc. and MicrogenDx. ALA is an advisor for Abbvie and Watershed Medical.

## Author summary

Urinary tract infections (UTI) are one of the most common infections in adults. Fear that these common localized infections will worsen to result in a serious systemic infection (urosepsis) drives significant antibiotic overprescribing, even though this complication is very rare. Early prediction of urosepsis is critical in improving the outcomes of those at high risk but could also reduce unnecessary antibiotics in those unlikely to benefit, but clinicians typically exhibit poor predictive judgement in distinguishing these populations. Machine learning (ML) techniques have the potential to aid healthcare professionals in identifying individuals at high risk for negative outcomes after an uncomplicated UTI. Numerous machine learning models were trained to predict 1) subsequent hospitalization after initially seeking care for a UTI and 2) urosepsis. Random forest models achieved meaningful predictive power for clinical risk stratification, noting that patient age, sex, and features of the microscopic urinalysis were the top predictive features associated with subsequent hospitalization and systemic infection. These models have the potential to assist clinicians in the identification of patients at high risk of complications after an outpatient presentation for UTI, which can improve both antibiotic stewardship and UTI outcomes.

## Introduction

Sepsis is a leading cause of death in United States hospitals, accounting for half of all hospital deaths [1]. Urosepsis, defined as sepsis progressing from urinary tract infection (UTI), comprises approximately 5–7% of all severe sepsis cases and has a mortality rate of 14% for community-acquired infections [2]. Early identification of urosepsis is crucial, as administering appropriate antibiotic treatment, providing supportive therapy, and identifying complicating factors, such as urinary obstruction, dramatically improves mortality rates [3]. Progression to urosepsis in adults is rare, however, occurring in only approximately 0.4% of all UTI [4].

Even though only a small proportion of patients will progress to systemic infection, antibiotic treatment of suspected UTI remains common practice; physicians routinely cite this fear as a reason for prescription of antibiotics even when they are not indicated [5]. As many as 75% of antibiotics prescribed for outpatient UTI are inappropriate, including unnecessary prescriptions, excessive duration of antibiotic therapy, and misuse of broad-spectrum antimicrobials [6–10]. The self-reported annual incidence of UTI in women is 12% with a lifetime prevalence of approximately 60% [11,12], making the antibiotic burden attributable to these inappropriate prescriptions a significant contributor to the worsening global crisis of antimicrobial resistance. UTI is one of the most common indications for antibiotics at outpatient visits to physician offices and emergency departments, comprising almost 25% of all outpatient antibiotics prescribed [13]. Randomized, placebo-controlled trials have shown, however, that antibiotic treatment for UTI offers only mildly faster symptomatic improvement compared to placebo in patients with urinary symptoms alone presenting in an outpatient setting

[14–17]. Indeed, the incidence of pyelonephritis in patients presenting to office-based settings is low and is not substantially different in individuals receiving antibiotics from those treated with supportive analgesics and hydration [18,19].

Given the rising threat of antimicrobial resistance, improved ability to identify the small subset of patients at risk of systemic progression would improve both care and antimicrobial stewardship, but can be challenging; early signs and symptoms in patients that progress to urosepsis are similar to uncomplicated patients whose condition self-resolves without intervention. Machine learning models offer a powerful approach to evaluate patients based on their predicted mortality or morbidity and to predict required resources for more efficient management [20]. Deriving disease subtypes from patient features that are available from electronic health records (EHRs) can help clinicians make better decisions for healthcare operation policies and guide next-generation personalized medicine [21,22].

While numerous studies have focused on the prediction of sepsis using machine learning, few focus specifically on the progression of UTI to urosepsis. Zhang et al. (2021) developed a sepsis risk prediction model for UTI patients using machine learning, achieving promising results but focusing primarily on inpatient populations and using a different set of predictive features. Existing research efforts have predominantly centered on specific clinical scenarios related to urosepsis, such as the differentiation of urosepsis from a urinary tract infection in inpatients, [23] predicting the risk of sepsis after flexible ureteroscopy [24], and antibiotic prescriptions for UTIs [25]. Notably, previous investigations have not explored the domain of forecasting urosepsis risk among a group of community-dwelling individuals initially diagnosed with a UTI, the bulk of individuals receiving antibiotic prescriptions. To date, only a single study [26] has attempted to study the association between antibiotic treatment for UTIs and severe outcomes in elderly patients in primary care settings. It is important to note, however, that this study used any hospital admission as a proxy for urosepsis, which is problematic; hospital admissions can result from a broad variety of medical conditions, of which urosepsis is a rare outcome of interest. The usage of a broad inclusion criteria can lead to incorrect inferences, especially for older patients, as hospital visits typically increase with age for numerous reasons. In fact, higher antibiotic burdens over time are associated with increased numbers of hospital admissions for a wide range of infections and antibiotic side effects [27–29].

This study sought to apply machine learning algorithms to predict the risk of urosepsis in patients presenting with a UTI in the outpatient setting, an area ripe for harnessing the potential of machine learning to improve patient care. We sought to derive a predictive algorithm for urosepsis risk, intentionally using a minimal number of features which could be obtained easily at the outpatient point-of-care to facilitate ease-of-use and ensure applicability across a range of care delivery settings.

## Methods

### Data

De-identified electronic health records were extracted from a single academic medical health system spanning primary to quaternary care. These data were deemed non-human subjects research by the local Institutional Review Board (IRB#21–001403). Inclusion criteria were a positive diagnosis of urinary tract infection indicated by ICD-10 code N30 and N93.0 and positive bacteria result via urinalysis in an ambulatory setting (primary care, urgent care, or emergency encounter). For these patients, we extracted demographic information, urinalysis findings, and any antibiotics prescribed for each instance of UTI. These features were indexed using the patient and encounter levels to account for multiple instances. Reencounters we defined as all clinical encounters within seven days of the initial UTI encounter. The reencounters were considered urosepsis-related if matching positive blood and urine cultures were identified with a sepsis ICD-10 code of A41, R78, or R65.

### Data preprocessing

The data matrix was constructed such that each row in this dataset corresponded to a unique outpatient encounter per patient, with the columns representing the various features of an encounter. Two predictor vectors were generated,

denoting whether or not within seven days of each outpatient encounter, (1) a patient was hospitalized for any cause, and/ or (2) the patient was hospitalized for a UTI-related complication. The extracted features included demographics such as patient age, BMI, ethnicity (White, Latinx, Black, Asian), sex, urinalysis features such as red blood cell (RBC), white blood cell (WBC), and squamous epithelial cell numbers per unit volume, and antibiotic prescriptions, including nitrofurantoin, fosfomycin, trimethoprim/sulfamethoxazole, cephalexin, ciprofloxacin, amoxicillin, amoxicillin/clavulanate, doxycycline, and levofloxacin. Preprocessing was done to extract and map categorical variables to numerical values, perform feature engineering to extract meaningful features, remove unformatted values and outliers lying outside three standard deviations and merge multiple antibiotics into binary columns. The features were standardized by removing the mean and scaling to unit variance before being input to the logistic regression model. Null imputation was performed in case of missing values. Demographics were dropped before computing overall model performance. The pipeline for model development is shown in Fig 1.

### Feature selection analysis

To address potential feature redundancy, we conducted a correlation analysis of all features in our dataset. We found out that feature 'UA_Micro' had the same value and was removed because of this redundancy. Next, our correlation analysis revealed minimal multicollinearity among our features (S1 Fig), with no feature pairs showing correlation coefficients above 0.8, a common threshold for identifying redundant features. Even the most related variables - urinalysis measurements (WBC and RBC counts) - showed correlations below problematic levels.

The lack of strong correlations among features, combined with the feature importance scores from our random forest models (Figs 2 and 3), suggests that each variable contributes unique information to the prediction task. While some

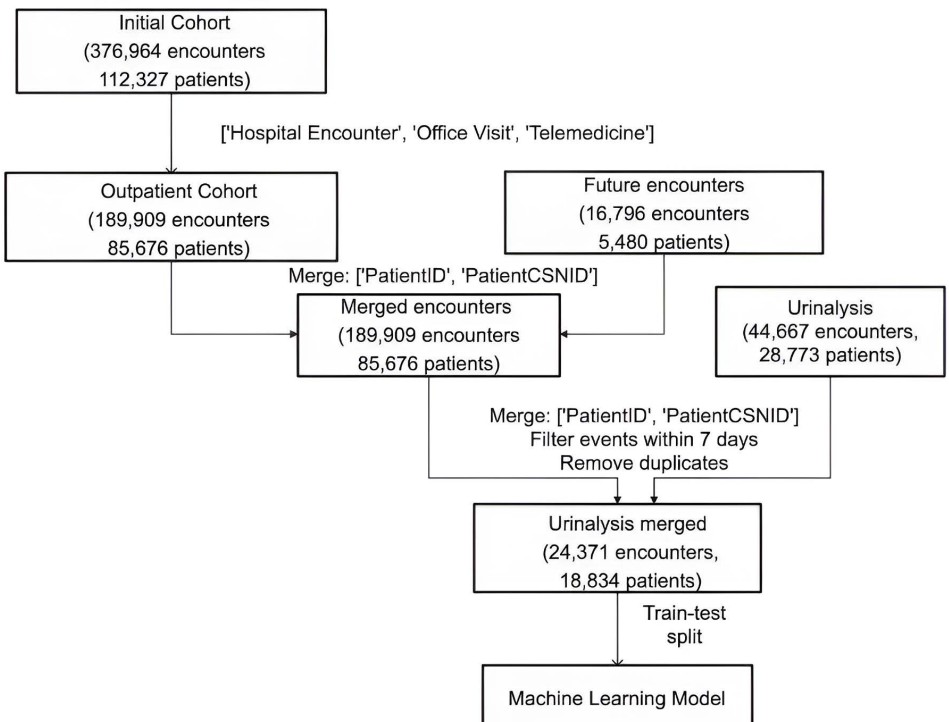

**Fig 1. Data preprocessing pipeline for input into machine learning model to predict hospitalization or urosepsis after outpatient presentation for urinary tract infection (UTI).**

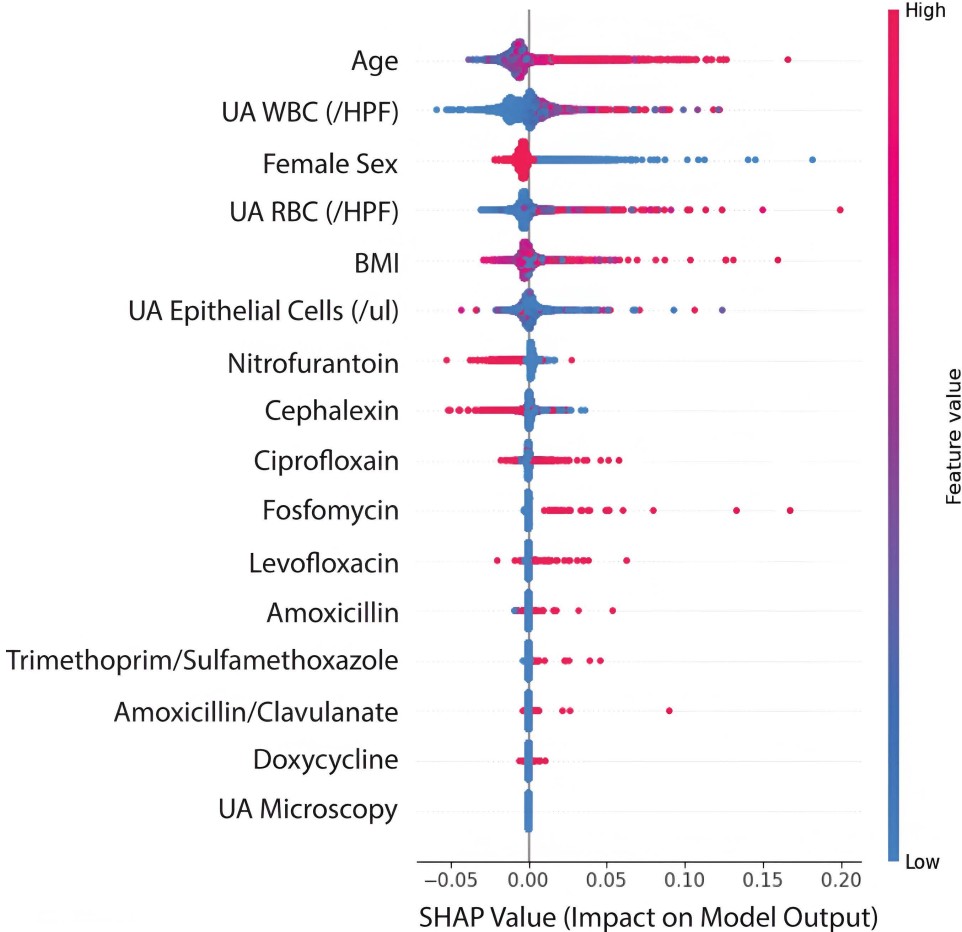

**Fig 2. *SHAP values from the random forest classifier for predicting rehospitalizations.*** A positive SHAP value denotes a positive impact on the model output, moving the prediction closer to +1 (no rehospitalization). A negative SHAP value denotes a negative impact on model output, moving the prediction closer to 0 (hospitalization). Each point on the plot represents an individual, and the color of the point denotes the feature value. Pink corresponds to higher values, while blue corresponds to lower values.

features (such as age and urinalysis findings) showed stronger predictive power, even lower-ranked features provided meaningful signal, supporting our decision to retain all features in the final model.

## Handling class imbalance

Our dataset exhibited significant class imbalance, with only 0.4% positive cases (urosepsis) versus 99.6% negative cases. To address this challenge, we evaluated multiple sampling strategies to improve model training:

1. Standard Random Forest: As a baseline machine learning approach, we trained the model on the original imbalanced data using the scikit-learn implementation with balanced class weights.

2. SMOTE (Synthetic Minority Over-sampling Technique): This approach generates synthetic examples of the minority class by interpolating between existing cases and their nearest neighbors, creating new training examples of the urosepsis class.

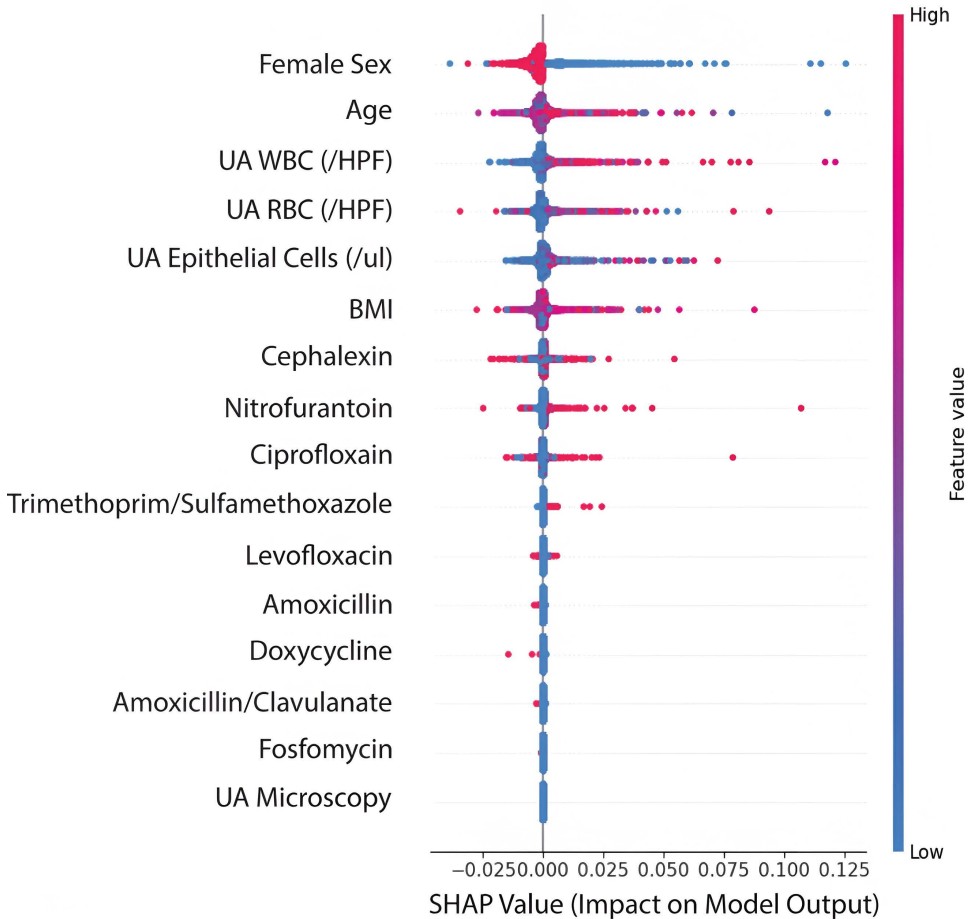

**Fig 3. *SHAP values from the random forest classifier for predicting Urosepsis.*** A positive SHAP value denotes a positive impact on the model output, moving the prediction closer to +1 (no urosepsis). A negative SHAP value denotes a negative impact on model output, moving the prediction closer to 0 (urosepsis). Each point on the plot represents an individual, and the color of the point denotes the feature value. Pink corresponds to higher values, while blue corresponds to lower values.

3. Random Oversampling: This method randomly duplicates minority class examples to achieve balanced classes, providing a simpler alternative to SMOTE's synthetic data generation.

4. Random Undersampling: This approach randomly removes majority class examples to achieve balance, testing whether a smaller, balanced dataset could improve prediction of the minority class.

5. Majority Classifier: For this method, each sample is predicted with the majority class. This helps explain why even the expected worst performing sampling strategy (under sampling with APR = 0.033) is still better than the baseline, since it is at least able to identify some positive cases, whereas the majority classifier identifies none.

For each strategy, we maintained the original class distribution in the test set to ensure unbiased evaluation. We compared these approaches using both precision-recall curves and ROC curves, with emphasis on precision-recall metrics given their greater sensitivity to performance on imbalanced datasets (S4 Fig).

## Machine learning analysis

This analysis sought to train a model that could predict the probability of an outpatient with a UTI having a reencounter due to urosepsis. We split our examination into two major analyses. First, we predicted the risk of a patient with a UTI being admitted to the hospital for any reason in a 7-day window. Next, we narrowed down our predictor variable to consider only reencounters that were demonstrated cases of urosepsis. We trained several models with different complexities and interpretability for two tasks, first, predicting all hospital reencounters, and second, predicting urosepsis. The dataset was randomly shuffled and split into training and test datasets in the ratio of 75:25. We implemented 4 machine learning models using the sklearn library:

1)  Logistic regression [30]: A linear model serving as our baseline, using the stochastic average gradient (SAG) solver for optimization. We chose SAG for its efficiency with larger datasets while maintaining good convergence properties.

2) Decision trees [31]: A non-linear model offering transparent decision rules. Hyperparameters were optimized through grid search, with final settings of maximum depth = 5 and minimum leaf size = 1 to prevent overfitting while maintaining predictive power.

3) Random forests [32]: An ensemble method combining multiple decision trees to improve robustness and performance. Grid search optimization led to settings of maximum depth = 15 and 100 trees, balancing model complexity with computational efficiency.

4) Neural Networks [33]: A multilayer perceptron classifier with two hidden layers (15 and 5 neurons respectively) and rectified linear unit (ReLU) activation functions. We used stochastic gradient descent for optimization with a learning rate (alpha) of 1e-5. This architecture was chosen for its ability to capture non-linear relationships while remaining computationally tractable for our dataset size.

Hyperparameter tuning for each of the models was done using a grid search over parameter space on the validation set. Confidence intervals were generated using bootstrapping by sampling with replacement [34]. The training and test data were randomly split 10 times, and models were fit on each of the shuffled datasets. These bootstrapped resamples were used to determine the 95% confidence interval. Model performance was assessed using both the receiver operating characteristic curve (ROC) and precision-recall curve (PRC), showing the tradeoff between positive predictive value and sensitivity for different thresholds. While our data was heavily imbalanced; there is literature showing the merits of using both metrics for model predictive performance [35].

## Results

### Data and data processing

The analysis dataset of 24,371 encounters from 18,834 patients was used as the input feature matrix in the machine learning models. Among these, 563 patients (2.9%) had a hospitalization within 7 days and 118 (0.4%) of these developed urosepsis (Table 1). While the final cohort had a substantially higher percentage of females, male gender was positively associated with urosepsis (p < 0.001). Increased urinary white and red blood cells were both associated with hospitalization, while only urinary WBC were associated with reencounters for urosepsis. The ages of both patients that were hospitalized (66.3) and patients with urosepsis (61.9) were typically older than the general population (52.6). Among individual antibiotics prescribed, only nitrofurantoin was negatively associated with urosepsis (p = 0.05).

### Predictive modeling

Logistic regression, decision tree, and random forests were evaluated as potential models. We used logistic regression as a baseline model and a decision tree for model interpretability. We used random forests as more complex models that have been proven to produce good results on unstructured data. The dataset was partitioned into 75% for training and

**Table 1. Outpatient cohort dataset summary of patients' feature information at encounter level, including demographic information and labs related to each episode of urinary tract infection (UTI). Significant p-values are highlighted in red.**

| Characteristics | Count (total) | Count (reencounter) | Count (urosepsis) | Rehospitalization | | Urosepsis | |
|---|---|---|---|---|---|---|---|
| | | | | Coefficient (Rehospitalization) | p-value | Coefficient (Urosepsis) | p-value |
| **Total** | 24,371 | 563 | 118 | | | | |
| **Sex** | | | | **-0.32** | **9e-26** | **-0.61** | **1.2e-18** |
| Male | 3,946 (16%) | 187 (33%) | 57 (48%) | | | | |
| Female | 20,423 (84%) | 375 (66%) | 61 (52%) | | | | |
| Other/ unknown | 2 (0.008%) | 1 (0.1%) | 0 (0%) | | | | |
| **Race/Ethnicity** | | | | | | | |
| Latin | 4214 (16.92%) | 102 (18.1%) | 31 (26.27%) | 0.11 | 0.14 | **0.02** | **0.007** |
| Asian | 2145 (8.80%) | 44 (78%) | 1 (0.84%) | 0.02 | 0.88 | -0.5 | 0.19 |
| White | 13111 (53.79%) | 305 (54.17%) | 70 (59.32%) | -0.02 | 0.72 | 0.01 | 0.57 |
| **Black** | 1626 (6.67%) | 54 (9.6%) | 3 (2.54%) | **0.12** | **0.03** | -0.47 | 0.07 |
| Other/ unknown | 3249 (13.82%) | 0 (0%) | 0 (0%) | | | | |
| **BMI** | | | | | | | |
| Mean(SD) | 25.94 (6.08) | 26.7 (8.57) | 26.92 (5.4) | -0.02 | 0.20 | 0.016 | 0.91 |
| **Age** | | | | | | | |
| **Mean(SD)** | 52.56 (22.91) | 66.26 (23.27) | 61.85 (22.02) | **0.58** | **1.22e-24** | **0.26** | **0.01** |
| **Labs** Mean(SD) | | | | | | | |
| **RBC (/HPF)** | 7.05 (20.51) | 13.289 (26.4) | 13.41(28.02) | **0.07** | **0.01** | -0.02 | 0.30 |
| **WBC (/HPF)** | 26.22 (41.08) | 49.42 (50) | 45.14 (53.58) | **0.33** | **2.55e-22** | **0.4** | **1.0e-07** |
| Squamous Epithelial Cells (/uL) | 15.15 (37.87) | 13.02 (36.72) | 10/28 (25.06) | 0.02 | 0.10 | -0.27 | 0.63 |
| **Antibiotics** | | | | | | | |
| **Nitrofurantoin** | 0.303 | 0.09 | 0.14 | **-0.48** | **0.06** | **-0.07** | **0.05** |
| **Fosfomycin** | 0.0094 | 0.04 | 0.05 | **0.07** | **0.003** | **0.08** | **7.3e-04** |
| Trimethoprim/ sulfamethoxazole | 0.102 | 0.008 | 0.016 | -0.02 | 0.66 | 0.04 | 0.59 |
| Cephalexin | 0.458 | 0.54 | 0.58 | -0.02 | 0.83 | 0.09 | 0.81 |
| Ciprofloxacin | 0.196 | 0.253 | 0.19 | -0.03 | 0.59 | -0.15 | 0.79 |
| Amoxicillin | 0.019 | 0.019 | 0 | -0.06 | 0.35 | -0.35 | 0.26 |
| Amoxicillin/ clavulanate | 0.0139 | 0.015 | 0 | -0.06 | 0.50 | -0.35 | 0.37 |
| Doxycycline | 0.0141 | 0.008 | 0 | -0.03 | 0.57 | -0.29 | 0.30 |
| Levofloxacin | 0.0271 | 0.062 | 0.048 | -0.04 | 0.46 | 0.02 | 0.74 |

25% for testing. ROC index, precision, and recall were used as evaluation metrics. A total of 21 features were used as input to the model, including demographic information such as age, sex, BMI, medication information, and laboratory findings (Table 1). We also calculated feature importance scores based on the mean decrease in Gini impurity using sklearn for decision trees and random forests to improve model interpretability (S1 Table).

For the task of predicting reencounters, all 4 models had comparable AUC, with logistic regression exhibiting the highest APR of 0.13 (as compared to a baseline APR of 0.004). We observed an improvement in model performance when we limited our predicted variable to urosepsis. Random forests were the best performing models with an AUC of 0.9, and an APR of 0.31, followed by decision trees, with AUC = 0.81 (as compared to a baseline of AUC of 0.5) (Fig 4). Neural

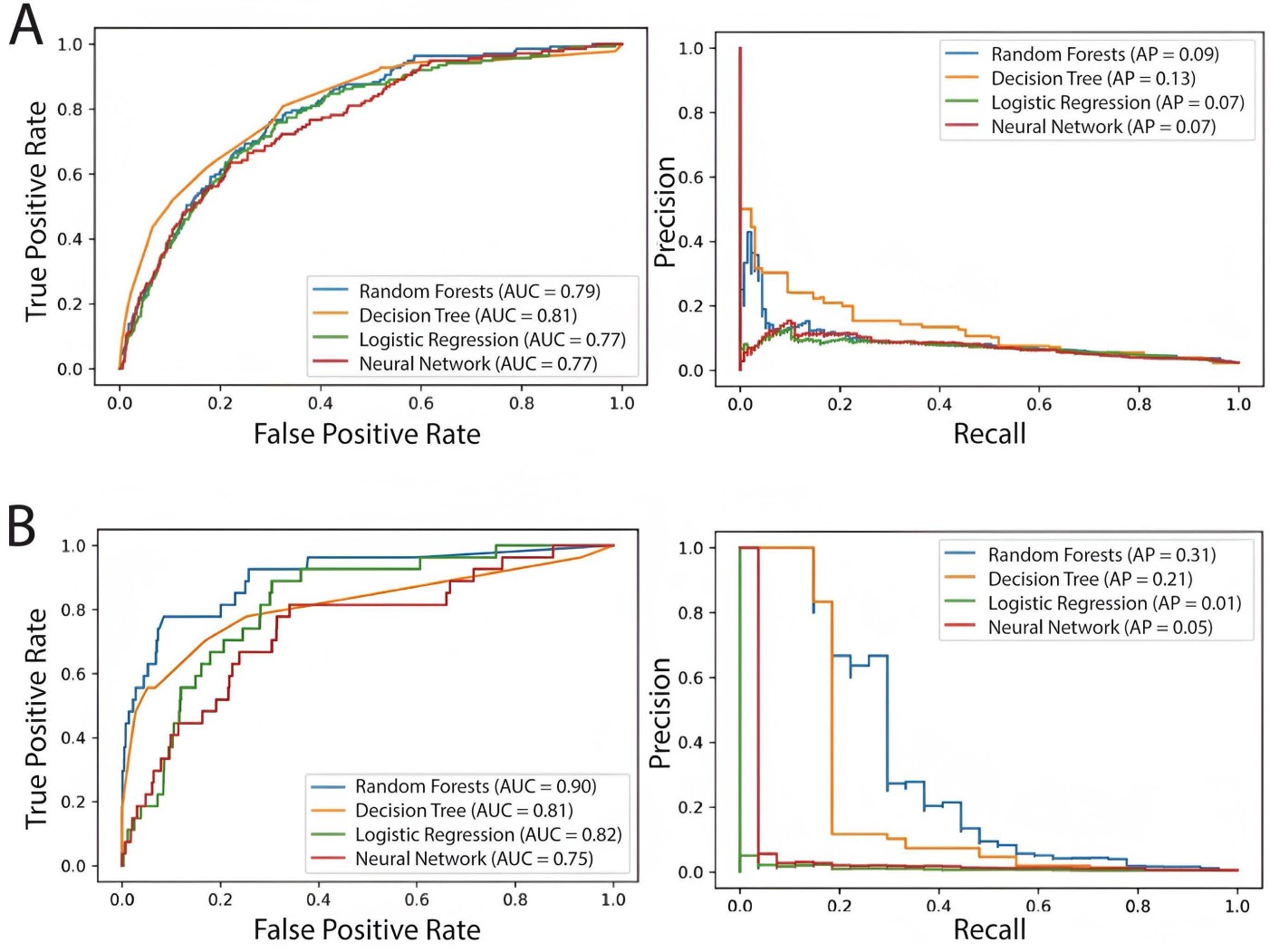

**Fig 4. *ROC and precision-recall curves for machine learning models.*** A) ROC and precision-recall curves for machine learning models predicting hospitalizations: for a 95% confidence interval, Decision trees achieved the highest AUC of 81%, random forests followed with 79%, logistic regression and neural networks with 77%. B) ROC and precision-recall curves for machine learning models predicting urosepsis. For a 95% confidence interval, Random Forests achieved the highest AUC of 90%, followed by Decision Trees, Logistic Regression and Neural Networks with AUCs of 81%, 82% and 75%, respectively.

networks were the lowest-performing models, with an AUC of 0.76. The most important feature in the prediction of both random forests and decision trees was the patient's age, followed by BMI and urinalysis WBC count (S1 and S2 Tables). For both models, since our dataset is severely imbalanced, with a 0.4% prevalence of Urosepsis, the APR values are much lower than the AUC values.

**Feature importance**

To analyze the local feature importance and the directionality of the effects, we generated beeswarm plots representing the Shapley values at an indiwvidual level for the random forest classifiers predicting hospitalization and urosepsis (Figs 2 and 3). We found patient age, sex, and urinalysis findings, including WBC and RBC counts, to be the most important

features for the prediction of both hospitalizations and urosepsis. An elevated white and red blood cell count lead to a positive impact on the model output, and so did being female. We observed an interesting trend in the antibiotics, with only nitrofurantoin having a negative value on the model output when prescribed. While fosfomycin was positively associated with both hospitalizations and urosepsis, none of the other individual antibiotics prescribed demonstrated a significant association with either reencounters or urosepsis specifically.

## Discussion

Outpatient presentations for UTI may account for as many as 6% of all outpatient medical visits [36]. Urosepsis is a rare, but serious complication of UTI for which early recognition and antibiotic administration can be essential [4]. However, pervasive antibiotic administration for all UTI does not appreciably decrease the risk of urosepsis [14–17] and may increase poor health outcomes with increased rates of hospitalization due to antibiotic adverse events, post-infectious complications, and subsequent multidrug-resistant infections [27–29]. Thus, identification of those patients at highest risk for septic progression of UTI might allow for targeted antibiotic administration to those in whom the benefits outweigh the risks and improve antimicrobial stewardship.

In this machine learning analysis, we generate machine learning models to predict the risk of sepsis following an outpatient encounter for UTI. We first considered the primary problem of predicting all possible reencounters at the hospital, and then the narrower problem of predicting microbially-defined urosepsis. Our machine learning models had varying performances, based on model complexity and interpretability of the results, but demonstrated that random forests were the best performing models, both in terms of ROC and model interpretability. The random forest model predicted urosepsis with an excellent AUC of 0.90 using only a minimum number of patient features, all of which could be obtained at point-of-care, such as patient age, BMI, and urinary WBC counts as the top important features. While neural networks are known to handle high dimensional datasets well in these circumstances its performance was limited due to small dataset sizes. Logistic regression and decision tree models exhibited intermediate performance, with baseline ROC of 0.81 and 0.82, respectively.

The performance characteristics of our models require careful interpretation in the context of our specific clinical prediction task. The relatively low accuracy of logistic regression can be attributed to the extreme class imbalance in our dataset (0.4% positive cases for urosepsis) and the likely non-linear relationships between clinical predictors and urosepsis development that logistic regression cannot adequately capture. This is supported by the superior performance of our random forest model. While this model achieved an excellent AUC of 0.90 and APR = 0.31, which represents meaningful improvement over the baseline APR of 0.004, and its performance characteristics align well with clinical needs in UTI management. The high precision means that when the model identifies a patient as high-risk for urosepsis, this prediction is likely to be correct, allowing for targeted intervention. While the lower recall indicates that some urosepsis cases may be missed, this limitation is acceptable within the current clinical context where most UTIs receive empiric antibiotic treatment.

A prime advantage of our analysis is that we have access to a unique dataset consisting of deidentified patient records spanning from primary to quaternary care, enabling us to track patient trajectories from primary care visit to hospitalization. This unique dataset, covering patients' journeys from their primary care doctor to hospital treatment, including the pertinent demographics, vital signs, laboratory testing and microbial culture results, allowed us to build several machine learning models demonstrating excellent performance predicting the risk of hospital reencounters, both overall and those specifically caused by urosepsis, after primary outpatient presentation for UTI.

Previous analyses of urosepsis risk after outpatient presentation for UTI have primarily used claims-based data detailing any-cause hospitalization or sepsis admission as a surrogate for urosepsis [26] as there is no specific ICD-10 code for urosepsis. As bacteriuria is common in hospitalized patients, which may lead to an inappropriate diagnosis of urosepsis in patients with an alternative cause for hospitalization, we specifically examined those patients in whom blood cultures

at hospitalization for sepsis reflected the same organism as that present on the urine culture obtained at the outpatient evaluation. Our data revealed that only the minority of hospital readmissions met the microbial definition of urosepsis, indicating that the majority of hospitalizations may be due to healthcare conditions unrelated to urosepsis; these data are in agreement with previous analyses demonstrating that all-cause hospitalizations were five-fold more common than urinary-infection related hospital admissions following outpatient presentation for UTI [37].

Thus, models examining hospital readmission are limited by the accuracy of the coding of the providers involved in each encounter, which is likely flawed. Accumulating data suggests that >60% of UTI diagnoses and ~75% of urosepsis diagnoses may be inaccurate [38–40], which implies that reencounters should not be used as a proxy for either incompletely-treated UTI or urosepsis. Indeed, using a better-defined predictor leads to a model with higher predictive performance, and more meaningful feature importance.

Given the high frequency of bacteriuria, particularly in older and institutionalized adults, any changes in overall health, such as waxing mental status or malaise/fatigue, can be mistakenly attributed to UTI despite numerous specialty society guidelines attempting to dispel this myth [41–43]. It is possible that in the context of misdiagnosis, other conditions that may be unrecognized or untreated due to this misdiagnosis may lead to the eventual hospitalization of the patient. It is also possible that antibiotic treatment itself may contribute to hospitalization [27–29]. As many as 20% of patients prescribed antibiotics will experience an adverse effect of the medication, from allergic reactions to dehydration from antibiotic-associated diarrhea or C. difficile colitis. In addition, frequent antibiotic use is associated with increased hospital admissions for infection-related complications in a dose-dependent manner [27]. Most of these readmissions are seen in the few days following antibiotic administration, and the rates of hospital admissions for infectious complications increase with larger cumulative antibiotic exposures. [28,29]. Future studies may need to account for cumulative antibiotic burden to further improve predictive modeling.

Cumulatively, these data support the use of the more stringent model in which we defined our predictor variable narrowly by the combination of concordant blood and urine microbial cultures and a sepsis ICD-10 code of A41, R78, and/or R65. Predictive ability as measured by AUC and APR was excellent using the more precise definition of urosepsis. While a high ROC may have been high due to the increased class imbalance, the APR, which is not affected by class imbalance, was still significantly above chance. We observed several changes from the more broadly defined readmission model in the top-most predictive features, as well as feature importance scores (S1 and S2 Figs). While the top few important features remained the same in predicting readmissions and urosepsis, it is important to note that the importance scores for antibiotics increase for the case of urosepsis. Nitrofurantoin, widely considered the first-line antibiotic for UTI due to its good penetration into the urinary tract and minimal collateral damage, remained the only antibiotic negatively associated with urosepsis risk. Interestingly, commonly prescribed broad-spectrum antibiotics, such as the fluoroquinolones ciprofloxacin and levofloxacin, were not associated with decreased risk of urosepsis, supporting previous data demonstrating that neither increased duration nor increased broad-spectrum agents are associated with improved outcomes in UTI [41,42]. Fosfomycin, which is also a first-line antibiotic for UTI [41,42] and the only oral agent with efficacy against extended-spectrum β-lactamase (ESBL)-producing E. coli, was associated with increased risk of urosepsis. As fosfomycin is typically effective in UTI treatment [44], we suspect that this association may be driven by the poor availability of fosfomycin in commercial pharmacies, frequently leading to delays in antibiotic administration. While potentially harmful when used indiscriminately, antibiotic usage plays an important role in the case of urosepsis, with delays in administration in these select cases leading to worse outcomes.

It is important to note, however, that only approximately 6% of all UTI-coded encounters had available urinalysis data for review. Of the initial dataset encompassing 377,000 visits for UTI, only 24,000 of these encounters had obtained the urinalysis data necessary for our analysis. This limitation significantly narrowed the scope of our dataset and underscores the challenges in data completeness for EHR-based studies. In addition, it highlights the need to establish expectations for the standardized evaluation of patients suspected of UTI. Most guidelines discussing the evaluation of patients with suspected UTI endorse obtaining urinary testing, typically including urinalysis and urine culture, at the time of evaluation; [41,42] these tests can be useful in ruling out UTI and ensuring prompt and appropriate treatment when indicated [45].

PLOS Digital Health

While educational interventions may be needed to improve UTI diagnostic accuracy, the availability of a risk calculator utilizing this information might also enhance guideline concordance of UTI evaluation as it would provide clinicians with a more tangible interpretation for these laboratory values that could better support clinical decision-making.

Future work could improve on these results by using a larger dataset with fewer missing values, imputing missing values in the existing dataset, and performing a chart review to randomly sample patients from each category. Refined models could include additional measurements and patient features, such as prior antibiotic burden, inclusion of comorbidities and medication history, and number of prior hospitalizations, to further improve model performance. It should be noted, however, that utilization of only a minimal number of factors, all of which could be obtained at the time of evaluation, provided excellent prediction of urosepsis hospitalization, which has the potential to dramatically improve both individual consequences of antibiotic overprescribing and global antimicrobial stewardship. Our study can help the scientific and medical community identify and prioritize high-risk patients, guide treatment, and improve clinical outcomes.

## Supporting information

**S1 Table. Features with their corresponding Gini feature importance scores for decision trees and random forests predicting hospitalizations after outpatient urinary tract infection (UTI).**
(DOCX)

**S2 Table. Features with their corresponding Gini feature importance scores for decision trees and random forests predicting urosepsis after outpatient urinary tract infection (UTI).**
(DOCX)

**S1 Fig. Feature Correlation Matrix.** Heatmap showing pairwise Pearson correlations between model features. Color intensity represents correlation strength, with darker colors indicating stronger correlations.
(TIFF)

**S2 Fig. Comparison of model performance across different sampling strategies for handling class imbalance.** Left: Precision-recall curves showing model performance with various sampling approaches. Oversampling achieved the highest average precision (AP = 0.20), followed by standard random forest (AP = 0.15), SMOTE (AP = 0.11), and under sampling (AP = 0.03), all outperforming the baseline (AP = 0.00). Right: Corresponding ROC curves demonstrating similar trends in model discrimination, with oversampling achieving the highest AUC (0.87), followed by SMOTE (0.89), standard random forest (0.85), and under sampling (0.84).
(EPS)

**S3 Fig. Summary plot showing cumulative importance for the prediction of hospitalization after outpatient urinary tract infection (UTI).**
(EPS)

**S4 Fig. Summary plot showing cumulative importance for the prediction of urosepsis after outpatient urinary tract infection (UTI).**
(EPS)

## Author contributions

**Conceptualization:** Georgina Dominique, Jeffrey N. Chiang, A. Lenore Ackerman.

**Data curation:** Varuni Sarwal, Georgina Dominique, Jeffrey N. Chiang.

**Formal analysis:** Varuni Sarwal, Nadav Rakocz, Jeffrey N. Chiang, A. Lenore Ackerman.

**Investigation:** Varuni Sarwal, Georgina Dominique, Jeffrey N. Chiang, A. Lenore Ackerman.

**Methodology:** Nadav Rakocz, Jeffrey N. Chiang, A. Lenore Ackerman.

**Project administration:** A. Lenore Ackerman.

**Software:** Nadav Rakocz.

**Supervision:** Jeffrey N. Chiang, A. Lenore Ackerman.

**Visualization:** Nadav Rakocz, Jeffrey N. Chiang.

**Writing – original draft:** Varuni Sarwal, Georgina Dominique, A. Lenore Ackerman.

**Writing – review & editing:** Varuni Sarwal, Georgina Dominique, Jeffrey N. Chiang, A. Lenore Ackerman.

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
