## [Decision Letter · Decision Letter 0]

PDIG-D-24-00227

Machine learning for the prediction of urosepsis using electronic health record data

PLOS Digital Health

Dear Author

Thank you for submitting your manuscript to PLOS Digital Health. After careful consideration, we feel that it has merit but does not fully meet PLOS Digital Health's publication criteria as it currently stands. Therefore, we invite you to submit a revised version of the manuscript that addresses the points raised during the review process.

Please submit your revised manuscript within 60 days Oct 15 2024 11:59PM. If you will need more time than this to complete your revisions, please reply to this message or contact the journal office at digitalhealth@plos.org. Please include the following items when submitting your revised manuscript:

We look forward to receiving your revised manuscript.

Kind regards,

Nadav Rappoport, Ph.D.

Academic Editor

PLOS Digital Health

Journal Requirements:

Additional Editor Comments (if provided):

Reviewers' comments:

Reviewer's Responses to Questions

**Comments to the Author**

1. Does this manuscript meet PLOS Digital Health’s publication criteria?

Reviewer #1: Yes

Reviewer #2: Yes

Reviewer #3: Yes

2. Has the statistical analysis been performed appropriately and rigorously?

Reviewer #1: Yes

Reviewer #2: Yes

Reviewer #3: Yes

3. Have the authors made all data underlying the findings in their manuscript fully available (please refer to the Data Availability Statement at the start of the manuscript PDF file)?

Reviewer #1: Yes

Reviewer #2: No

Reviewer #3: Yes

4. Is the manuscript presented in an intelligible fashion and written in standard English?

PLOS Digital Health does not copyedit accepted manuscripts, so the language in submitted articles must be clear, correct, and unambiguous. Any typographical or grammatical errors should be corrected at revision, so please note any specific errors here.

Reviewer #1: Yes

Reviewer #2: Yes

Reviewer #3: Yes

Reviewer #1: The study developed a machine learning model for predicting the risk of hospitalization and urosepsis in outpatient urinary tract infection patients. Standardized disease coding with ICD-10 was utilized for the included conditions, and the model achieved satisfactory predictive performance. However, there are several limitations in this research:

1. In 2016, sepsis was defined by new diagnostic criteria, and the R65 code used in the study stands for systemic inflammatory response syndrome. The urosepsis cases discussed in the study actually encompass both patients with positive blood culture for sepsis and those with systemic inflammatory response syndrome, indicating the presence of confounding factors among the urosepsis patients in this study.

2. Although the study assessed the impact of 16 features on the model, it did not select from the 16 features included in the model. If a subset of optimal features could be screened out from the 16 features, that is, by removing some redundant features, the model's performance might be improved.

Reviewer #2: Please see the attached reviewer report for the full comments.

Major comments:

1. Abstract: Our models demonstrated high predictive performance with an area under the ROC curve

(AUC) of 79.5% AUC and an area under the precision-recall curve (APR) of 13% APR for

reencounters" - PR AUC of 13% sounds not high and ROC is less relevant for imbalance

binary classification as mentioned in the Method section. Can you explain the claim that

these results are high predictive performance? In addition, please clarify what are the

baselines that the results with high predictive performance are compared to.

2. Introduction: There is a previous work by Zhang et al. (2021) that appears highly relevant to this

research but is not mentioned in the related work section of the introduction.

3. Please justify the choice of these three machine learning models: logistic regression, decision

tree, and random forest. For example, explain why XGBoost was not used.

4. Have the authors considered any approaches to address the data imbalance? in a scenario of

significant data imbalance, exploring approaches that address imbalanced datasets is highly

relevant and essential to include in the manuscript.

5. Missing baselines: consider adding a random classifier and an always positive or negative

classifier as baselines for comparison.

6. The reproducibility of this work is limited since the dataset is private and the code is not available currently. Therefore, consider applying your method to other datasets as well, especially public datasets under basic

restrictions such as MIMIC-IV or eICU-CRD if possible. Additionally, reporting the results

of another dataset can enhance the generalizability of this work.

7. Discussion, paragraph two: "but demonstrated that random forests were the best performing models, both

in terms of ROC and model interpretability" - In a task where the data is highly imbalanced,

ROC indeed is not relevant. Moreover, I think ROC performance can be misleading for that

task given the extreme data imbalance. In light of this, I think it is better to remove this

metric from the results. APR is more relevant, and reporting the precision and recall of the

positive label (at a specific threshold) might also be interesting for real-world scenarios.

Please see the attached reviewer report for further comments.

Reviewer #3: The manuscript reports of a novel, systematic study of a problem that was not well address before. ML-based prediction of urosepsis is important can could have a significant impact on patients' lives. We believe this is a very promising study and we encourage the authors to further improve it following the comments below.

The authors correctly suggest that their dataset is significantly imbalanced (0.4% of the class of interest). AS we see it, the authors do not do enough in their study in response to this imbalance. One response it the use of precision-recall curve. This response is good to have a better metric to measure the quality of the results, and we are happy for the authors to have used it. Another response by the authors is data resampling, however resampling has a limited effect in the case of 0.4% vs. 99.6%. There should likely be both oversampling of the minority class and under-sampling of the majority class. Also, some synthetic data can be added to the minority class. In addition, the small number of minority class records might lead to overfitting. The authors did not check this. Or maybe there is a possibility to get more real data?

To summarize the imbalance problem - the authors need to carefully refer to all of the risks associated with this imbalance and suggest what they do to address each, as needed.

In the methods section, the authors suggest that they use logistic regression, decision trees, and random forests, for their ML analysis. But once we move to the results section, we find out that neural networks were used too. This is inconsistent. Also, if ANNs were used, more details about them are needed: which type of ANN was used? Why that type? More details on the other models could be useful too, to facilitate study repeatability.

We believe the above comments require additional work but can be performed and will improve the manuscript significantly.

**Do you want your identity to be public for this peer review?** For information about this choice, including consent withdrawal, please see our Privacy Policy

Reviewer #1: No

Reviewer #2: Yes: Ofir Ben Shoham

Reviewer #3: No

---

## [Decision Letter · Decision Letter 1]

Response to Reviewers
Revised Manuscript with Track Changes
Manuscript
**Additional Editor Comments (if provided):**

Kindly implement the recommendation by the Reviewer 3. There will be no re-review after that and the manuscript can go straight into acceptance and production mode.

With best regards,

**Reviewers' Comments:**

**Comments to the Author**

Reviewer #1: All comments have been addressed

Reviewer #2: All comments have been addressed

Reviewer #3: All comments have been addressed

publication criteria?

Reviewer #1: Yes

Reviewer #2: Yes

Reviewer #3: Yes

3. Has the statistical analysis been performed appropriately and rigorously?

Reviewer #1: Yes

Reviewer #2: Yes

Reviewer #3: Yes

4. Have the authors made all data underlying the findings in their manuscript fully available (please refer to the Data Availability Statement at the start of the manuscript PDF file)?

Reviewer #1: Yes

Reviewer #2: No

Reviewer #3: No

5. Is the manuscript presented in an intelligible fashion and written in standard English?

Reviewer #1: Yes

Reviewer #2: Yes

Reviewer #3: Yes

Reviewer #1: (No Response)

Reviewer #2: I thank the authors for addressing my concerns and comments. I recommend accepting the revised manuscript.

Reviewer #3: I am happy with the authors' responses to my comments - they fully address them. I have comments regarding Figure S4: 1. the fonts used are too small; 2) the legend in the left sub-figure covers hides parts of the curves, and therefore should be repositioned.

**Do you want your identity to be public for this peer review?** For information about this choice, including consent withdrawal, please see our Privacy Policy

Reviewer #1: No

Reviewer #2: **Yes: ** Ofir Ben Shoham

Reviewer #3: No

**Figure resubmission:****Reproducibility:** To enhance the reproducibility of your results, we recommend that authors of applicable studies deposit laboratory protocols in protocols.io, where a protocol can be assigned its own identifier (DOI) such that it can be cited independently in the future. Additionally, PLOS ONE offers an option to publish peer-reviewed clinical study protocols. Read more information on sharing protocols at https://plos.org/protocols?utm_medium=editorial-email&utm_source=authorletters&utm_campaign=protocols

---

## [Decision Letter · Decision Letter 2]

Machine learning for the prediction of urosepsis using electronic health record data

PDIG-D-24-00227R2

Dear Dr. Ackerman,

We are pleased to inform you that your manuscript 'Machine learning for the prediction of urosepsis using electronic health record data' has been provisionally accepted for publication in PLOS Digital Health.

Best regards,

Martin G Frasch

Section Editor

PLOS Digital Health

**Additional Editor Comments (if provided):**

**Reviewer Comments (if any, and for reference):**

Reviewer's Responses to Questions

**Comments to the Author**

Reviewer #3: All comments have been addressed

publication criteria?

Reviewer #3: Yes

3. Has the statistical analysis been performed appropriately and rigorously?

Reviewer #3: Yes

4. Have the authors made all data underlying the findings in their manuscript fully available (please refer to the Data Availability Statement at the start of the manuscript PDF file)?

Reviewer #3: Yes

5. Is the manuscript presented in an intelligible fashion and written in standard English?

Reviewer #3: Yes

Reviewer #3: My comments from earlier revisions were fully addressed.

**Do you want your identity to be public for this peer review?** For information about this choice, including consent withdrawal, please see our Privacy Policy

Reviewer #3: No
